# Challenges in the Application of African Swine Fever Vaccines in Asia

**DOI:** 10.3390/ani14172473

**Published:** 2024-08-25

**Authors:** Agathe Auer, Giovanni Cattoli, Pawin Padungtod, Charles E. Lamien, Yooni Oh, Sarah Jayme, Andriy Rozstalnyy

**Affiliations:** 1Joint FAO/IAEA Center, 2444 Seibersdorf, Austria; 2Food and Agriculture Organization of the United Nations (FAO), 00153 Rome, Italy; 3Istituto Zooprofilattico Sperimentale delle Venezie, 35020 Padua, Italy; 4Food and Agriculture Organization of the United Nations (FAO), Representation in Vietnam, Hanoi 11112, Vietnam; 5Food and Agriculture Organization of the United Nations (FAO), Regional Office for Asia and the Pacific, Bangkok 10200, Thailand

**Keywords:** vaccine safety, DIVA vaccination, ASF control in endemic countries

## Abstract

**Simple Summary:**

African swine fever (ASF) presents a significant challenge in Asia, where outbreaks have devastated the pork industry and threatened food security due to high mortality rates in pigs. The disease, caused by the ASF virus genotype II, has spread rapidly across the region since its emergence in China in 2018. This paper discusses the introduction and implications of vaccines such as NAVET-ASFVAC and AVAC ASF Live in Vietnam, emphasizing the necessity for rigorous testing and regulatory oversight. Despite the potential use of vaccines to control the disease, concerns about the safety of live attenuated vaccines (LAVs), including their ability to revert to virulence or create new recombinant strains, highlight the complexity of ASF management. Effective vaccine strategies, alongside strict biosecurity measures, and rapid diagnostics are essential to mitigate the economic and social impacts of ASF and ensure the stability of pig populations in Asia.

**Abstract:**

This paper explores the significance of quality vaccines in managing ASF in Asia, where it poses a substantial threat to the pork industry. It emphasizes the risks associated with substandard vaccines, including the emergence of new virus strains that complicate disease control. Highlighting recent advancements in vaccine deployment in Vietnam, the paper calls for rigorous testing and regulations to guarantee vaccine effectiveness and safety. The authors advocate for the implementation of vaccines with the inclusion of differentiating infected from vaccinated animals (DIVA), which enhances disease management strategies in both endemic and non-endemic regions. The conclusion underscores the necessity of stringent standards in vaccine development and strict adherence to regulatory guidelines to ensure successful ASF management and maintain public trust in the vaccines.

## 1. Introduction to ASF and Current Disease Situation in Asia

African swine fever virus (ASFV) genotype II, following its emergence and rapid spread in China in 2018, poses a critical threat due to its high fatality rate in infected pigs (domestic and wild). Since then, outbreaks in 19 countries within this region have been reported to the World Animal Health Information System (WAHIS) as of July 2024 (Figure 1). The social, health, and economic impacts on communities due to ASF are severe. For instance, in China, retail pork prices rose by 78%, heavily impacting consumers. It is estimated that by the end of 2019, the national pig herd in China was reduced by half as a consequence of ASFV genotype II outbreaks [1]. Similarly, in Vietnam, a quarter of pig populations succumbed to the diseases or had to be culled during the first year after ASF’s arrival. Low- and middle-income countries suffer severe socio-economic losses, in addition to threats to food security and livelihoods [2]. These impacts are further complicated by the difficulty in assessing the true global impact due to frequent under-reporting in endemic countries, where sick pigs continue to be traded to recover some of the losses [3].

ASFV is a stable double-stranded DNA virus that is classified within the family Asfarviridae and the genus Asfivirus. Genotyping of ASFV involves amplifying and sequencing the variable 3′ end of the B646L gene [4]. This gene encodes for p72, the primary capsid protein, which is crucial for tracing and identifying different strains of ASFV [5,6]. More than 20 genotypes of ASFV have been identified in Africa; however, only genotypes I and II have been detected outside the continent. ASFV genotype II is currently responsible for global ASF outbreaks, while genotype I, largely eradicated from regions outside of Africa, was recently detected in China [7]. Notably, the last reported cases of genotype I in Europe occurred in Sardinia, with one detection in a wild boar in 2019 and one in a domestic pig in 2018 [8,9,10]. The genotype I viruses identified in China to date do not resemble those found in Sardinia, but are more closely related to two attenuated viruses obtained in Portugal in 1968 and 1988 [7].

ASFV primarily infects pig macrophages, altering signaling pathways and disrupting the expression of genes associated with both innate and acquired immunity. These cells are also widely utilized in vitro for virus detection and research, given their susceptibility to ASFV infection [4,11]. Transmission occurs through direct contact and bodily excretions such as blood, urine, feces, and saliva, although airborne transmission is unlikely. ASFV can also be transmitted by Ornithodoros ticks. If Ornithodoros ticks and pigs co-exist in an ecological niche situation, this presents a significant risk in virus persistence in the field [12,13]. Infected animals are most contagious during the late stages of the disease, when clinical signs are evident and often only post-mortem. However, viral shedding in oronasal and lachrymal secretions, urine, and feces can occur as early as 1 to 7 days post-infection (dpi), with shedding from the oral cavity occurring prior to systemic dissemination of the virus. Even low amounts of the virus are sufficient to cause efficient in vivo infection, highlighting the importance of early detection and control measures [14]. Blood is particularly significant in ASFV transmission, and like other arboviruses, ASFV can also be transmitted through ticks.

## 2. Vaccine as a Tool to Control Disease and Limit Economic Losses—The Example of Vietnam

To combat the threat of ASF, Vietnam has taken proactive measures by licensing two vaccines: NAVET-ASFVAC, manufactured by NAVETCO, and AVAC ASF Live, produced by AVAC, in June 2022 for field trials. Subsequently, on 24 July 2023, the Ministry of Agriculture and Rural Development (MARD) approved the nationwide use of these vaccines, emphasizing the need for vaccine quality and safety standards in the field. NAVET-ASFVAC is based on ASFV-G-ΔI177L, demonstrating stability and attenuation following five passages in domestic pigs [15]. AVAC ASF Live is based on the ASFV-G-∆MGF strain, featuring six deletions and propagated in Diep’s macrophage cell (DMAC) line. Moreover, the Philippines recently announced that its ASFV vaccination campaign with AVAC will commence in 2024 (Table 1) [16]. However, farmers in Vietnam are cautious in using these vaccines, echoing concerns similar to those faced during vaccine trials in the Philippines [2]. The long-term safety and efficacy data of these two LAVs in the field are not publicly available, raising concerns about potential reversion to virulence. This cautious approach highlights the critical need for stringent vaccine regulation and thorough testing to ensure safety and efficacy in the field. Indeed, the World Organization for Animal Health (WOAH) warned that current ASF vaccines need more testing [17]. At the same time, vaccines have been exported to the Philippines, Indonesia, Malaysia, India, and Myanmar for limited trials [2,18]. Issues such as shedding of vaccine virus, vertical and horizontal transmission, immunogenicity to various field strains, reversion to virulence, potential for recombination with field virus, and post-vaccine complications, among others, are unresolved issues in vaccine development, which are particularly critical in LAVs. These challenges are addressed in the new draft WOAH standard, which advocates for the development and evaluation of ASF vaccine candidates to ensure they meet regulatory approval criteria [19]. Table 1 shows the distribution of ASFV vaccine doses by NAVETCO and AVAC up to June 2024, with NAVETCO distributing a total of 667,000 doses and AVAC distributing 3,601,710 doses across Vietnam, the Dominican Republic, the Philippines, and Nigeria.

## 3. The Types of Vaccines Applied in Asia, Field Vaccine-Related Viruses, and Their Impact on Disease Surveillance and Control

Recent research has identified the emergence of attenuated genotype I ASFV strains, HeN/ZZ-P1/21 and SD/DY-I/21, in China. These strains are characterized by reduced virulence, but high transmissibility, leading to chronic conditions such as necrotic skin lesions and joint swelling [21]. Phylogenetic analysis indicates that these strains are closely related to isolated genotype I viruses, namely, strains NH/P68 and OURT88/3, which were isolated in Portugal [22]. In 2023, reports suggested potential recombination of these genotype I strains with pandemic genotype II viruses, creating a highly virulent recombinant strain [23,24]. This recombinant virus exhibits high lethality and rapid transmission akin to genotype II, complicating the ASF landscape further. Given the absence of evidence for the persistent circulation of NH/P68 and OURT88/3in other endemic areas, its emergence is particularly alarming. A plausible hypothesis is the introduction of this genotype I strain through illicit vaccine trials aiming to develop new live attenuated or genetically modified vaccines. This genotype I/II recombinant might have been created naturally in the field as well, given the co-circulation of both genotypes. Thus, both laboratory and field origins of the recombinant are possible, with no evidence that can conclusively exclude one or the other. However, no recombinant mosaic virus has so far been detected outside Asia or in Africa, where more than 20 genotypes are simultaneously circulating. The illicit application of vaccines further complicates the epidemiological landscape, exemplified by the case of ASFV-GUS-Vietnam, identified in Northern Vietnam in 2021. It is important to clarify that ASFV-GUS-Vietnam is the same virus as the AVAC vaccine virus (ASFV-G-∆MGF). [25]. Notably, this genetically modified LAV initially caused only mild clinical signs, but displayed increased virulence in subsequent trials [26]. While it provided complete protection against a homologous challenge, the increased virulence raises serious safety concerns and underscores the necessity for stringent regulation and careful oversight in deploying live attenuated recombinant ASF vaccines in field settings. The occurrence of clinical signs with the disease patterns observed after 7 days post-inoculation with GUS-Vietnam in a recent study from 2024 contrasts with a previous study from 2015, where no clinical signs were observed when ASFV-G-∆MGF was inoculated into domestic pigs [26,27]. Notably, ASFV-GUS-Vietnam was found prior to the release of licensed ASF vaccines in Vietnam. These examples show that the illicit use of improper vaccines can not only spread the disease but also facilitate the emergence of new variants in the regions, in addition to the risk of recombination between different ASFV genotypes.

In the case of ASF, a pig infected with the highly virulent strain ASFV genotype II typically develops severe hemorrhagic disease, often resulting in death within a few days. A targeted vaccine against this genotype should prevent the animal from dying. However, the presence of recombinant viral strains alters the disease pattern, potentially undermining efforts like partial culling [28]. The slow spread of ASF through a farm (low contagiousness) can usually be leveraged to implement a partial or selective culling strategy effectively, but this approach may not work if the disease pattern changes due to strains such as ASFV-GUS-Vietnam. In several Asian countries, partial culling has been adopted as a strategy to manage ASF’s spread while minimizing economic losses. For example, Vietnam officially adopted partial culling in July 2019, which helped preserve over 50% of the pig population and extended control measures by only eight days, although the long-term impact on livelihoods remains uncertain [29]. Similarly, South Africa successfully eliminated ASF in specific areas using this method [30]. Other countries, including Laos, Cambodia, India, and China, have also implemented partial culling strategies [17]. Understanding the high-risk period (HRP)—the duration that the virus is present before detection—is essential to prevent spread and minimize losses in partial culling. Early detection is crucial for controlling secondary outbreaks and managing the spread through animal movements, contaminated products, and other vectors. However, challenges arise with the introduction of uncontrolled vaccines or illicit LAVs, which can disrupt existing control strategies. Furthermore, the absence of a serological DIVA vaccine complicates the rapid identification of infected versus vaccinated animals, necessitating whole-genome sequencing for definitive differentiation. Although Brake (2022) argues that in endemic regions where ASF is widespread, DIVA vaccines might not be essential since vaccination can still reduce disease spread by diminishing viral shedding and alleviating clinical symptoms [31], DIVA vaccines remain crucial for effective disease management in both endemic and non-endemic areas. They help accurately distinguish between vaccinated and naturally infected animals, which can be isolated and culled. The licensed ASFV-G-∆I177L vaccine and other candidates contain molecular markers that facilitate the monitoring of vaccinated populations [32]. Given these complexities, countries must weigh the expected vaccine efficacy, achievable population coverage, and likely compliance when considering ASF vaccination programs.

## 4. The Importance of Development and Application of Safe, Effective Vaccines

The situation in Asia with ASF mirrors challenges faced in the recent past with other diseases, such as lumpy skin disease (LSD). In Asia, the LSDV vaccine (Neethling strain) encountered several critical safety and effectiveness issues, including contamination of vaccine batches with wild-type LSDV and goat-pox virus (GTPV) [33]. This contamination introduced more strains into the vaccinated population, including recombinant virulent strains. These recombinant strains exhibited unpredictable behavior and increased virulence, complicating clinical outcomes in vaccinated animals. Furthermore, the presence of both vaccine and wild-type strains, as well as recombinant strains, led to significant diagnostic confusion, making it difficult to differentiate between vaccinated and naturally infected animals. These challenges were documented in various studies highlighting the complexity and risks associated with using contaminated or recombinant vaccine strains [33,34,35]. Moreover, molecular characterization revealed that the Bangladeshi LSDV strains were genetically distinct from contemporary field strains. Indeed, whole-genome sequencing showed that Bangladeshi LSDV strains had unique genetic markers distinguishing them from other contemporary strains. These strains shared similarities with historical isolates from Kenya and other parts of Africa from the 1950s, suggesting an older strain’s reintroduction into the region [36,37]. Pre-analysis of vaccines could have shown that LAVs against LSDV are not safe for use in animals. Similarly, for the genotype I ASFV strain, as observed in the abovementioned case of LSDV, there is a possibility that the strain was introduced through vaccines. The provided examples show that vaccines not rigorously tested can even introduce new variants into regions that were previously free of such strains.

## 5. Addressing Current Vaccine Challenges

Building on the critical importance of developing and applying safe, effective vaccines, as discussed, it is imperative to address the ongoing challenges and potential risks associated with vaccine quality and safety. In Asia, the experiences with diseases like ASF and LSD underscore the necessity for rigorous testing, regulation, and the strategic development of vaccines to control high-impact animal diseases effectively. Ensuring vaccine integrity not only supports disease management but also prevents the exacerbation of disease spread due to inadequate vaccine quality. Recently developed ASF vaccines can potentially be used to limit the economic losses and disease spread [38]. However, it is essential to examine the context of vaccine use and disease impact in Asia. The importance of these ASF vaccines is underscored by the critical role pigs play in food security and the economy. WOAH (2023) [17] warns against the use of non-compliant and poor-quality vaccines, which, as described in this communication, may have caused recombination between the vaccine virus and either the field virus or another virus used in other vaccines. This recombination can result in new viral strains capable of causing acute, chronic, or persistent infections.

Ongoing global research focuses on developing live attenuated, subunit, inactivated, and vector-based vaccines, which are still under extensive testing and not yet widely available. Traditional inactivated vaccines, including chemically inactivated or irradiated ones, have shown limited protective effects against ASF [39,40]. Research has revealed reduced viral loads and an immune response, but these vaccines did not prevent severe symptoms, leading to the euthanasia of vaccinated and control animals. A multi-epitope subunit vaccine using immunoinformatics, analyzing 18,858 proteins from 100 ASFV proteomes, has shown high antigenicity and immunogenicity, but remains untested in animals [40]. Challenges include identifying protective antigens, defining immune correlates, and scaling production in cell culture [30,41]. LAV candidates for ASF, focusing on the p72 genotype II strain, use naturally attenuated and recombinant viruses with gene deletions [31,42,43]. Pigs vaccinated with ASFV-G-ΔMGF exhibited mild, transient symptoms without significant virulence reversion. However, there are ongoing concerns about the genetic stability of the virus due to the “ΔMGFnV” mutation, which involves deletions and duplications of ASFV genes [44]. Notably, the development of mutant strains and the occurrence of mild symptoms and prolonged viremia were evident after just five passages. This raises significant concerns about field applications where thousands of pigs would be vaccinated, potentially leading to more severe issues [44]. Thus, it cannot be ruled out that vaccinated animals are shedding the virus, plus there is the concern of reversion to virulence in LAVs.

## 6. Conclusions

In conclusion, the examples described in this paper demonstrate the role that vaccines can play in controlling animal diseases and limiting economic losses, while simultaneously highlighting the negative impacts that can arise from improper vaccine application. The types of vaccines applied in Asia and the emergence of field vaccine-related viruses highlight the need for stringent biosecurity measures and comprehensive surveillance. The importance of developing and applying safe, effective vaccines cannot be overstated, as illustrated by WOAH’s concerns and the challenges faced in similar situations, such as LSD in Asia. The use of an unsafe vaccine in endemic areas in Africa where multiple genotypes circulate would likely increase the challenges faced by poor people whose pigs provide a much-needed source of income. Ensuring vaccine quality and safety through rigorous testing and regulation is essential for effective disease control and management. To improve trust and support decision-making regarding ASF vaccines, a standard for long-term field monitoring of vaccine impact is needed, which vaccine developers and manufacturers can follow. Additionally, the establishment of independent bodies to evaluate vaccine efficacy and safety in Asia is essential.

## Figures and Tables

**Figure 1 animals-14-02473-f001:**
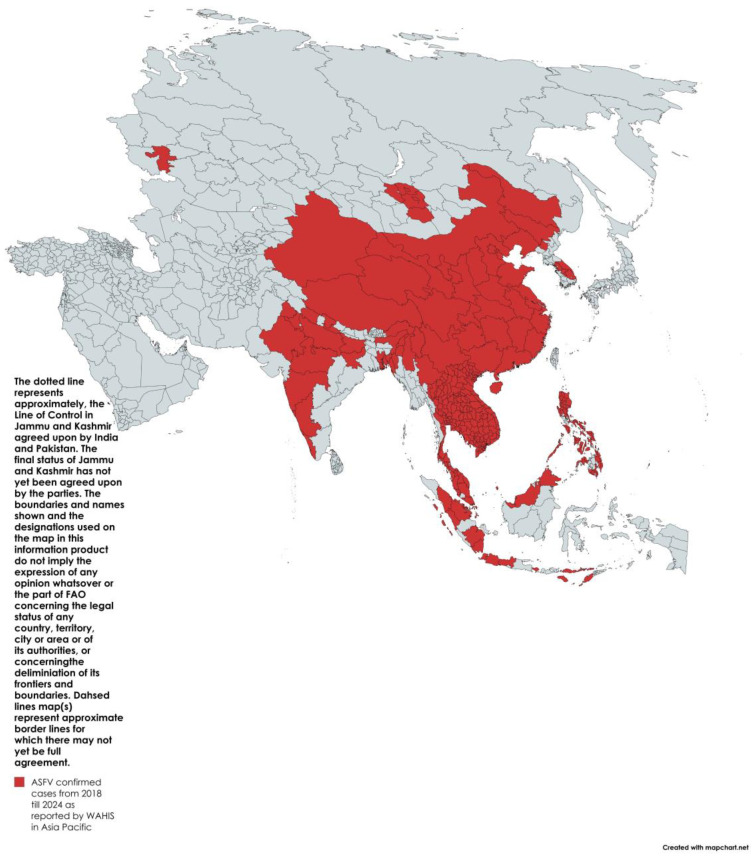
Distribution of confirmed ASF cases in Asia–Pacific from 2018 to 2024 as reported by WAHIS (red).

**Table 1 animals-14-02473-t001:** ASFV vaccines distributed (doses) till June 2024 [20].

	NAVETCO	AVAC
Vietnam	660,000	3,296,710
Dominican Republic	7000	-
Philippines	-	300,000
Nigeria	-	5000
Total	667,000	3,601,710

## Data Availability

Data available upon request.

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
