# Peer review of "Challenges in the Application of African Swine Fever Vaccines in Asia"

_animals, 2024, doi:10.3390/ani14172473_

Round 1

Reviewer 1 Report

Comments and Suggestions for Authors

This is a very relevant "commentary" to the so far unregulated  procedures on the  development

and use of LAVs  against ASF.  Argumentation is solidly

based on relevant data. Criticism to relevant

matters on the lack of efficiency an safety in currently develloped vaccines, is rationally and strongly presented.

Overall this manuscript contains a very relevant analysis of important issues on the development and use of vaccines against ASF.

Author Response

Dear Reviewer 1,

Thank you for your thorough review and constructive feedback on our manuscript. Below, and in the revised manuscript, we address each of your comments:

  1. Line 54 – add reference to ASFV classification

The missing reference was added (Reference 4: Galindo, I.; Alonso, C. African Swine Fever Virus: A Review. Viruses 2017, 9, 103. https://doi.org/10.3390/v9050103).

  1. Line 63 – “ASFV primarily infects macrophages” – it is true but it might be interesting to relate to the “in vivo” infection because these cells are also the best well-known targets for “in vitro” detection etc. add reference

The comment was included as follows: These cells are also widely utilized in vitro for virus detection and research, given their susceptibility to ASFV infection.

Reference was added (Reference 11: Sánchez, E.G.; Pérez-Núñez, D.; Revilla, Y. Mechanisms of Entry and Endosomal Pathway of African Swine Fever Virus. Vaccines 2017, 5, 42. https://doi.org/10.3390/vaccines5040042).

  1. Line 67-68 – “Infected animals are most contagious during the late stages of the disease when clinical signs are evident and often only post-mortem”- Please add reference. Descriptions by others like: “Viral shedding in oronasal and lachrymal secretions, urine and faeces occurs 1 to 7 dpi….shedding from the oral cavity occurs prior to systemic dissemination of the virus” (in: Sanchez-Cordón, PJ, Vidaña, B et al.; Pathology of African swine fever. Understanding and combating African swine fever. DOI 10.3920/978-90-8686-910-7_4, ) and because even low amounts of virus cause efficient “in vivo” infection, it would be interesting to justify the above-mentioned argument.

The comment was addressed in the text as follows: However, viral shedding in oronasal and lachrymal secretions, urine, and feces can occur as early as 1 to 7 days post-infection (dpi), with shedding from the oral cavity occurring prior to systemic dissemination of the virus. Even low amounts of the virus are sufficient to cause efficient in vivo infection, highlighting the importance of early detection and control measures.

Reference was added accordingly.

  1. Line 108 - .. “of this 35-year old strain” – which strain?

This related to NH/P68 and OURT88/3 and is now clearly stated in the text.

  1. Line 182-183 - ..”Like what has been observed for the genotype I ASFV strain…” probably to add something like “the above mentioned…”

The sentence was revised accordingly to: Similarly, for the genotype I ASFV strain, as observed in the above-mentioned case of LSDV, there is a possibility that the strain was introduced through vaccines.

  1. Line 201 – “… or another vaccine virus” ? meaning virus used in other vaccines?

The sentence was revised accordingly.

We believe the revisions made have strengthened our manuscript.

Sincerely,

Dr. Auer

Reviewer 2 Report

Comments and Suggestions for Authors
  1. The title is broad and no-specific. Vietnam is as specific country that was taken as an example for ASF, regarding the disease importance and vaccines application (Lines, 17-18). Therefore, the country or “Vietnam” and the “vaccine” could be added to the title.   
  2. A table compares between different ASF vaccines that were used worldwide could be added.
  3. The abbreviations all over the article should be revised. The abbreviation should be written as full words and then followed by (the standard abbreviation) that should be mentioned all over the article.
  4. Lines 210 and 21, references such as “Simbulan et al. (2024)” and “Deutschmann et al. (2023)” should be mentioned in a correct way.
  5. In conclusion, try to not add references.
  6. A specific title for the “Conclusions” and another one for “Future Perspectives” could be added for more specification.

Best wishes

Author Response

Dear Reviewer 2,

Thank you for your thorough review. We have carefully addressed each of your comments as follows:

  1. The title is broad and non-specific. Vietnam is a specific country that was taken as an example for ASF, regarding the disease importance and vaccines application (Lines 17-18). Therefore, the country or “Vietnam” and the “vaccine” could be added to the title.

The title was changed to: "Challenges in the Application of African Swine Fever Vaccines in Asia." The title change from "Potential Role of Vaccines in the Prevention and Control of African Swine Fever" to "Challenges in the Application of African Swine Fever Vaccines in Asia" was necessary to better reflect the manuscript's focus on the broader regional implications rather than limiting it to a single country. Including "Asia" instead of just "Vietnam" is more appropriate because it acknowledges that the challenges and strategies discussed are relevant to multiple countries within the region where these vaccines are being distributed and used, providing a more accurate and comprehensive scope of the issue.

  1. A table comparing different ASF vaccines that were used worldwide could be added.

A table showing ASFV vaccines distributed in doses until June 2024 was added (Table 1). This data was presented by the Department of Animal Health at the Conference on Sustainable Pig Farming Development, held by the Ministry of Agriculture and Rural Development on 14th August 2024 in Ha Noi. The table provides a clear overview of vaccine distribution across various countries, highlighting the scale and reach of these vaccines.

  1. The abbreviations all over the article should be revised. The abbreviation should be written as full words and then followed by (the standard abbreviation) that should be mentioned all over the article.

This was changed accordingly.

  1. Lines 210 and 21, references such as “Simbulan et al. (2024)” and “Deutschmann et al. (2023)” should be mentioned in a correct way.

References are cited in the text correctly now.

  1. /6. In conclusion, try to not add references. A specific title for the “Conclusions” and another one for “Future Perspectives” could be added for more specification.

This was changed as suggested, so that no references are in the conclusion part.

Thank you once again for your feedback. We believe these revisions have strengthened the manuscript, and we appreciate your efforts in helping to improve the quality of the work.

Best regards,

Dr. Auer

Reviewer 3 Report

Comments and Suggestions for Authors

Auer et al. present a timely review of the safety of modified live virus African swine fever vaccines, and the text covers the main issues. The authors do need to check their references carefully as I found issues with two of them (see below).

I suggest the following minor alterations to improve accuracy and clarity.

Line 43, suggest changing to “reduced by half due to ASF virus genotype II outbreaks” to “reduced by half as a consequence of ASF” to avoid the reader drawing the conclusion that reduction in the Chinese herd was caused by ASFV itself. Rather than the overall effect on the pig industry.

Figure 1. This should show regions of Russia reporting ASF as it borders the Pacific and is in Asia. Legend is in title case and remove clause “indicating districts where ASF in Present” at the end as ASF may be not be present in those regions now, or when the reader reads the commentary.

Line 53 to Line 62 and Line 115. One of the genotypes has been proven to be a lab error (https://doi.org/10.1128/mra.00067-24). Suggest rewriting this sentence to avoid a mentioning a specific number of genotypes.

The genotype I viruses identified in China to date do not resemble those identified in Sardinia, but are more closely related to two attenuated viruses, which are genetically very closely related, obtained in Portugal in 1968 and 1988. The sentence as written might lead the reader to think the genotype I viruses found in China are directly related to those found in Sardinia which is definitely not the case. This section needs to be carefully rewritten.

Line 66, please clarify what is meant by droplet, assume airborne?

Line 66, Ornithodoros ticks were also involved to the epidemiology of ASFV in Spain and Portugal (https://doi.org/10.1371/journal.pone.0020383). If Ornithodoros and pigs co-exist in an ecological niche situation this presents a significant risk in virus persistence in the field.

Line 81. My understanding is that the Philippines have trialled the AVAC vaccine, but not the NAVET one.

Line 114 to Line 115. There is some suggestion of recombination within specific genetic loci (https://doi.org/10.3201/eid1704.101283, https://doi.org/10.1038/s41598-020-75377-y), but the authors are correct to state that no mosaic virus like the one the genotype I/II hybrid has been identified as yet.

Lines 117 to Lines 129. The ASFV-Gus-Vietnam virus is the same as the AVAC vaccine.

Reference 15, the link goes to a blank page.

Reference 26, the author list doesn’t match that at the doi.

Author Response

Dear Reviewer 3,

Thank you for your thorough review and feedback. We have carefully addressed each of your suggestions as follows:

  1. Line 43, suggest changing to “reduced by half due to ASF virus genotype II outbreaks” to “reduced by half as a consequence of ASF” to avoid the reader drawing the conclusion that reduction in the Chinese herd was caused by ASFV itself. Rather than the overall effect on the pig industry.
    This was changed accordingly.
  2. Figure 1. This should show regions of Russia reporting ASF as it borders the Pacific and is in Asia. Legend is in title case and remove clause “indicating districts where ASF in Present” at the end as ASF may be not be present in those regions now, or when the reader reads the commentary.
    Figure and legend have been revised.
  3. Line 53 to Line 62 and Line 115. One of the genotypes has been proven to be a lab error (https://doi.org/10.1128/mra.00067-24). Suggest rewriting this sentence to avoid a mentioning a specific number of genotypes. The genotype I viruses identified in China to date do not resemble those identified in Sardinia, but are more closely related to two attenuated viruses, which are genetically very closely related, obtained in Portugal in 1968 and 1988. The sentence as written might lead the reader to think the genotype I viruses found in China are directly related to those found in Sardinia which is definitely not the case. This section needs to be carefully rewritten.
    We would like to thank you for the provided paper. We have also clarified this part to avoid giving the false impression that the genotype I found in China is related to genotype I in Sardinia.
  4. Line 66, please clarify what is meant by droplet, assume airborne?
    Changed to airborne.
  5. Line 66, Ornithodoros ticks were also involved to the epidemiology of ASFV in Spain and Portugal (https://doi.org/10.1371/journal.pone.0020383). If Ornithodoros and pigs co-exist in an ecological niche situation this presents a significant risk in virus persistence in the field.
    We have added this information to the text accordingly.
  6. Line 81. My understanding is that the Philippines have trialled the AVAC vaccine, but not the NAVET one.
    This is clearly stated in Table 1 and in the text.
  7. Line 114 to Line 115. There is some suggestion of recombination within specific genetic loci (https://doi.org/10.3201/eid1704.101283, https://doi.org/10.1038/s41598-020-75377-y), but the authors are correct to state that no mosaic virus like the one the genotype I/II hybrid has been identified as yet.
    The sentence has been clarified.
  8. Lines 117 to Lines 129. The ASFV-Gus-Vietnam virus is the same as the AVAC vaccine.
    This is now clearly stated in the revised manuscript.
  9. Reference 15, the link goes to a blank page.
    We have double-checked this reference and revised it to: "Press Release - Villar, Cynthia: Opening remarks during the public hearing on the proliferation of the unauthorized sale of African Swine Fever (ASF) in the market while the same is still undergoing trial and the role of the BAI and the FDA with regards to importation of vaccine (senate.gov.ph)."
  10. Reference 26, the author list doesn’t match that at the doi.
    This has been corrected.

Thank you again for your comments, which have greatly contributed to improving the accuracy and clarity of our manuscript.

Best regards,

Dr. Auer